# The *Aspergillus nidulans* velvet domain containing transcription factor VeA is shuttled from cytoplasm into nucleus during vegetative growth and stays there for sexual development, but has to return into cytoplasm for asexual development

Anja Strohdiek[1], Anna M. Köhler [1], Rebekka Harting[1], Helena Stupperich[1], Jennifer Gerke [1], Emmanouil Bastakis [1], Piotr Neumann[2], Yasar L. Ahmed [2], Ralf Ficner[2], Gerhard H. Braus [1]*

**1** Department of Molecular Microbiology and Genetics and Göttingen Center for Molecular Biosciences (GZMB), Institute of Microbiology and Genetics, University of Göttingen, Göttingen, Germany,
**2** Department of Molecular Structural Biology and Göttingen Center for Molecular Biosciences (GZMB), Institute for Microbiology and Genetics, University of Göttingen, Göttingen, Germany

* gbraus@gwdg.de

## Abstract

Survival of multicellular organisms requires the coordinated interplay between networks regulating gene expression and controlled intracellular transport of respective regulators. Velvet domain proteins are fungal transcription factors, which form various heterodimers and play key roles in controlling early developmental decisions towards more either asexual or sexual differentiation. VeA is the central subunit of the trimeric velvet complex VelB-VeA-LaeA, which links transcriptional to epigenetic control for the coordination of fungal developmental programs to specific secondary metabolite synthesis. Nuclear localization of the VeA bridging factor is carefully controlled in fungi. In this work we demonstrate that VeA carries three nuclear localization signals NLS1, NLS2 and NLS3, which all contribute to nuclear import. We show that VeA has an additional nuclear export sequence (NES) which provides a shuttle function to allow the cell to relocate VeA to the cytoplasm. VeA is nuclear during vegetative growth, but has to be exported from the nucleus to allow and promote asexual development. In contrast, progression of the sexual pathway requires continuous nuclear VeA localization. Our work shows that an accurate nuclear import and export control of velvet proteins is further connected to specific stability control mechanism as prerequisites for fungal development and secondary metabolism. These results illustrate the various complex mutual dependencies of velvet regulatory proteins for coordinating fungal development and secondary metabolism.

of the Creative Commons Attribution License,
which permits unrestricted use, distribution,
and reproduction in any medium, provided the
original author and source are credited.

**Data availability statement:** The data for the
crystal structure is free available in the RCSB
Protein Data Bank accessable with the code
9I2S. All other data are found in the manuscript
and Supporting Information.

**Funding:** This work was supported by funding
from the Deutsche Forschungsgemeinschaft
(DFG: https://www.dfg.de) to GHB (BR1502/19-
1 and IRTG PRoTECT). LCMS for metabo-
lite analysis was funded by the Deutsche
Forschungsgemeinschaft (INST 186/1287-1
FUGG to GHB). We acknowledge support by
the Open Access Publication Funds of the
University of Göttingen. The funders had
no role in study design, data collection and
analysis, decision to publish, or preparation of
the manuscript.

**Competing interests:** The authors have
declared that no competing interests exist.

## Author summary

A controlled interplay of networks that control gene expression and the intracellu-
lar transport of their respective regulators is essential for survival of multicellular
organisms. The velvet DNA-binding and dimerization domain is highly conserved
in fungi. It shares the same fold as the Rel homology domain of the mammalian
NF–κB immune cell response regulator. This includes conserved amino acid
residues required for DNA binding. Fungal velvet domain proteins like VeA are
transcription factors. They promote either more asexual or sexual differentia-
tion. VeA is nuclear during vegetative growth and also for the sexual pathway. In
contrast, VeA has to be exported from the nucleus to allow and promote asexual
development. This complex localization control requires the interplay between
three signals for nuclear import as well one for nuclear export within VeA. This
is further combined with VeA stability control. Relocalization of VeA allows the
fungus to react to changes of environmental conditions and to adapt its devel-
opment and secondary metabolism accordingly. This tightly controlled nuclear
import as well as export for coordinating fungal development and secondary
metabolism illustrates the importance of various complex mutual dependencies
of velvet regulatory proteins in fungi.

## Introduction

The filamentous fungus *Aspergillus nidulans* belongs to the Ascomycota [1,2].
The life cycle of this fungus starts with asco- or conidiospore germination, which is
followed by a vegetative phase before reaching developmental competence after
12–20 h [3,4]. It can either reproduce sexually or asexually. The fate of the develop-
mental program is determined by the reception of specific environmental and endog-
enous signals [3]. Light and oxygen induce asexual development, whereas darkness,
low oxygen and elevated $CO_2$ levels are sufficient to trigger sexual development [5].
Conidiophores are formed during asexual development harboring asexual conidia [6].
Cleistothecia are the sexual fruiting bodies, developed in the sexual cycle and are
surrounded by Hülle cells forming nests [7]. Hülle cells protect and sustain the matur-
ing cleistothecia especially by production of secondary metabolites like xanthones
[8,9]. The *A. nidulans* life cycle is well studied and numerous tools for genetic manip-
ulation are available. This allows molecular gene targeting studies to understand the
function of genes and their respective proteins like transcription factors.

*A. nidulans* development is regulated on molecular level by the velvet domain fam-
ily transcription factors, which are highly conserved through fungi [10,11]. The velvet
domain family consists of VeA (velvet A), VelB (velvet-like protein B), VelC (velvet C)
and VosA (viability of spores A). All members have a velvet domain of approximately
200 amino acids which is conserved in the fungal kingdom and has structural similar-
ity with the Rel homology domain (RHD) of the mammalian transcription factor NF–
κB, suggesting a common ancestor [12]. The velvet domain in VelB is interrupted by
a 99 amino acid intrinsically disordered domain. VeA has a continuous velvet domain

located in the N-terminal region and contains a previously predicted nuclear localization signal (NLS) [13] and nuclear export signal (NES). Localization sequences are required to transport proteins between cytoplasm and nucleus, which provides a regulatory layer for fungal development and secondary metabolism [14].

Velvet proteins form heterodimers which have different roles in fungal development [10]. VeA is localized in the cytoplasm when *A. nidulans* is exposed to light and found in the nucleus in darkness [11]. The α-importin KapA (Karyopherin A) is suggested to shuttle the VeA-VelB heterodimer into the nucleus where it induces sexual development [15]. A loss of either the *veA* or *velB* gene leads to a defect in sexual development with complete loss of cleistothecia production [11]. The velvet complex consists of VeA interacting N-terminally with VelB and C-terminally with the methyltransferase LaeA (loss of *aflR* expression A) (VelB-VeA-LaeA). This complex links fungal development with secondary metabolism, combining transcriptional and epigenetic control of gene expression [15]. The interaction of VelB with VeAs velvet domain at the same residues which contain a putative NLS, let to the hypothesis that additional NLS might be required for velvet complex cellular shuttling.

This study investigates the interplay between one NES and three NLS signals to control cellular trafficking of VeA. Different VeA variants were expressed carrying inactivated NLS and NES motifs. These strains were used to study VeA cellular localization, its interaction with import proteins such as KapA and the role of these signal sequences for velvet mediated fungal development and secondary metabolism. Here we show that VeA NLS motifs contribute to nuclear exclusion of VeA during illumination and thereby support asexual development. The NES regulates VeA nuclear export, which is mandatory for a coordinated fungal development and secondary metabolism. The NLS signals contribute to the nuclear import of VeA during the sexual program. VeA NLS motifs are required for adequate development of reproductive structures in combination with the associated secondary metabolism.

## Results

### Predicted *A. nidulans* VeA nuclear localization (NLS) and nuclear export (NES) sequences

VeA acts with its N-terminal part of the velvet domain as bridging factor for VelB and C-terminally to LaeA resulting in the formation of the trimeric velvet complex VelB-VeA-LaeA [15]. The cNLS mapper tool [16] predicted in the N-terminus of VeA the bipartite NLS1 (amino acid residues aa 28–45) [13] and in the C-terminus additional monopartite NLS2 (aa 507–516) and bipartite NLS3 (aa 537–563) (Fig 1A). Furthermore, the LocNES mapper tool [17] predicted an NES (aa 188–194) in the velvet domain (Fig 1A).

N-terminal VeA is required for VeA-VelB interaction and contains the bipartite NLS1, which promotes nuclear import and subsequently sexual development in darkness [15]. We determined the crystal structure of the N-terminal *A. nidulans* VeA part (aa 1–224) including the velvet domain [19] and combined it with data obtained by AlphaFold1 [20] and PyMOL (PyMOL Molecular Graphics System, Version 2.0 Schrödinger, LLC). This revealed that the NLS1 in the N-terminus of VeA is embedded in the velvet fold. This surrounding could be sterically exclusive and might either only allow binding of importin KapA or of velvet domain protein VelB [15]. In contrast to NLS1, the modelled C-terminal domain of the VeA protein suggests that NLS2 and NLS3 are more accessible for importin binding. The different signal sequences were therefore analyzed by codon exchanges within all potential nuclear import or export sequences for the indicated amino acid residues to assess their molecular cellular function in VeA localization (Fig 1A).

These results provide the possibility that nuclear transport of VeA is regulated by a complex interplay of potential NLS and NES motifs. An additional regulatory layer might be the formation of dimers or multimers, a process that possibly also interferes with nuclear transport of VeA.

### NLS1-3 cooperate in VeA nuclear import and are required to exclude the protein from the nucleus in light when asexual development is promoted

Localization of VeA-GFP NLS variants as well as VeA-GFP wild type was analyzed with cultures incubated in light on tilted agar slides to induce asexual development and in darkness for sexual development. The localization of the VeA variants during vegetative growth was analyzed by using cultures that were grown in liquid with illumination. The fluorescence

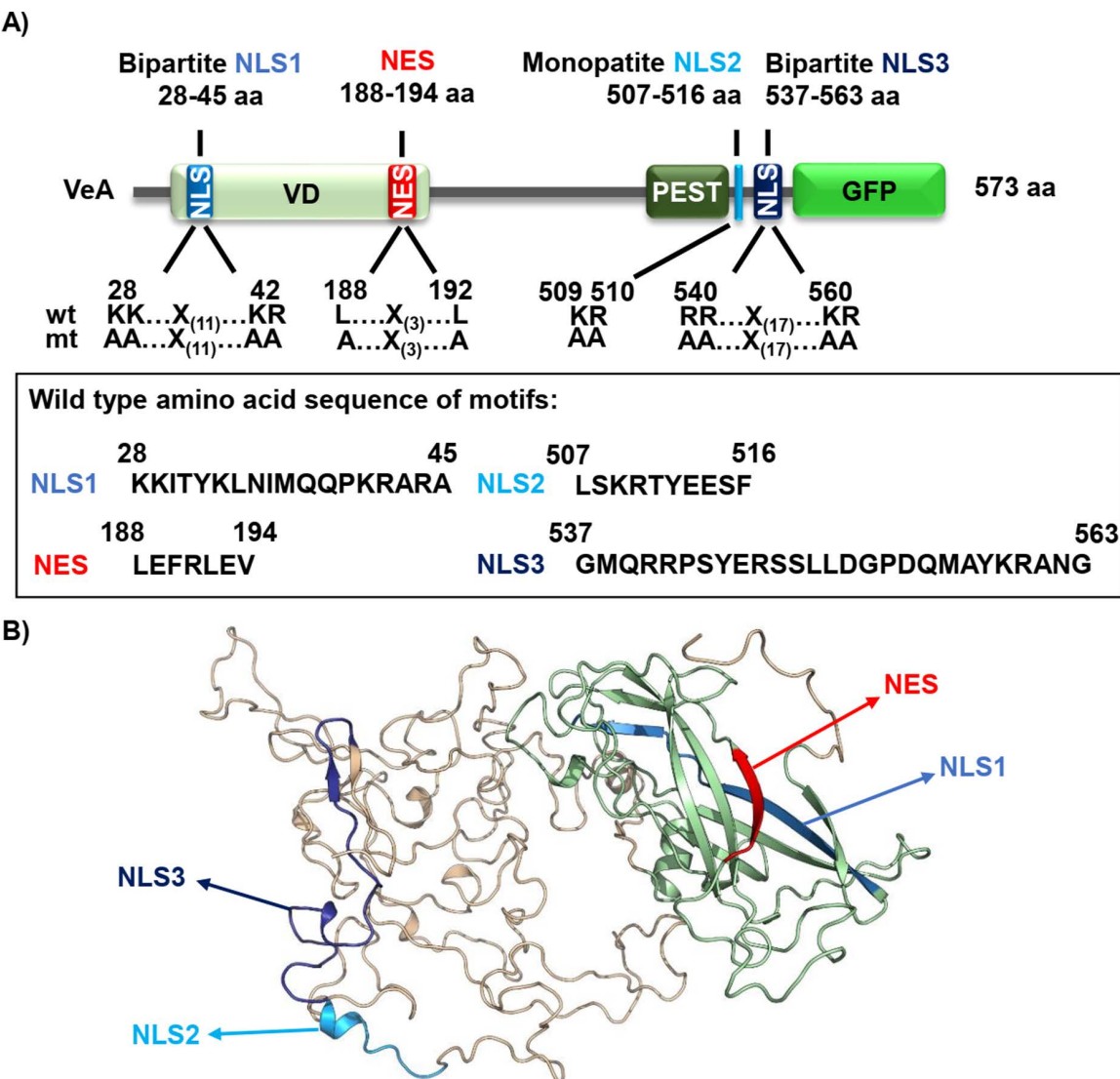

**Fig 1. A prediction revealed NLS1 and NES in VeAs velvet domain and C-terminally exposed NLS2/ NLS3. (A)** NLS prediction with cNLS mapper [18] revealed the bipartite NLS1 (blue) in the velvet domain (light green) as well as the monopartite NLS2 (turquoise) and bipartite NLS3 (dark blue) in the C-terminal part of the VeA protein. In addition, a potential nuclear export signal (red) is located at the end of the velvet domain [17]. The full wild type amino acid sequences of each motifs are indicated in the box. Functional studies of all NLS and the NES sequences were conducted by codon exchanges for the indicated amino acids (wt: wild type, mt: mutant). **(B)** Crystal structure of VeA velvet domain (green) [19] combined with the predicted structure of the C-terminus of VeA using AlphaFold1 [20] and PyMOL (PyMOL Molecular Graphics System, Version 2.0 Schrödinger, LLC.). The NLS and NES are highlighted as in A.

microscopy revealed that full length VeA-GFP is predominantly localized in the cytoplasm during asexual development inducing illumination conditions (Fig 2). In contrast, VeA is located in the nucleus and stimulates the sexual program in darkness (Fig 2B and 2C) [15].

   VeA variants defective in any of the three NLS show an enhanced nuclear accumulation of VeA to different extents during illumination, but a similar distribution as VeA wild type during darkness. VeA*NLS1*NLS2-GFP is distributed in the cell like wild type protein under both conditions. However, VeA*NLS1*NLS2*NLS3-GFP leads to an increased

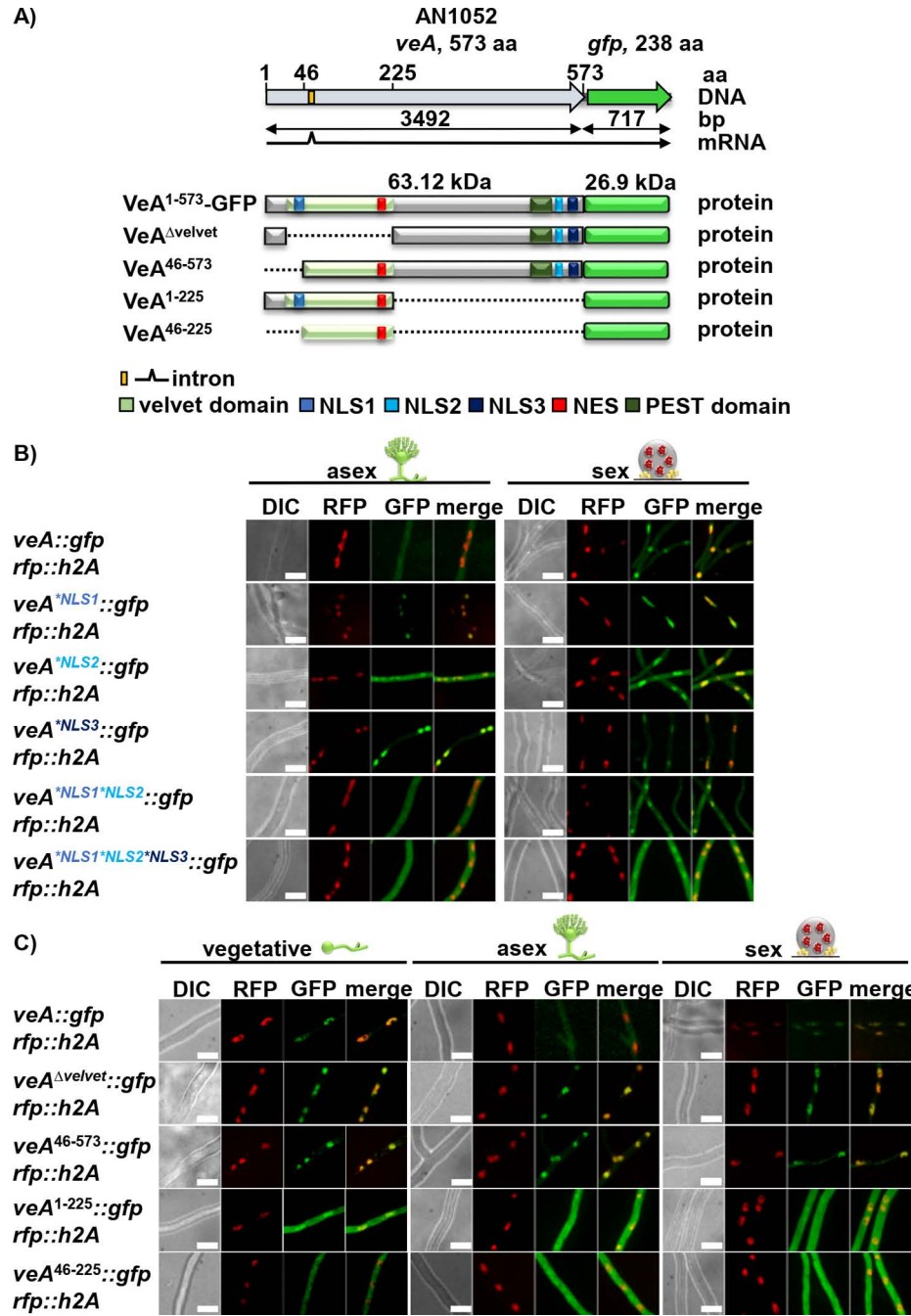

**Fig 2. VeA NLS are additionally required to exclude VeA from the nucleus to promote asexual development. (A)** Scheme of VeA-GFP with its domains and predicted NES and NLS motifs as well as different truncated versions of VeA-GFP. VeA$^{\Delta velvet}$-GFP is missing the velvet domain. VeA$^{46-573}$-GFP is missing the first 45 amino acids together with the NLS1. VeA$^{1-225}$-GFP is missing the C-terminus of VeA with the NLS2 and NLS3 motifs. VeA$^{46-225}$-GFP lacks all three NLS motifs. **(B)** Fluorescence images of strains expressing VeA-GFP and VeA-GFP NLS versions with amino acid substitutions grown on agar slant surfaces after 18 h illumination for promoting asexual (condiophore = light) or darkness (cleistothecia = darkness) for sexual development. Nuclei are visualized with RFP-H2A (size bars: 10 µm). **C)** Fluorescence microscopy was performed with VeA-GFP, VeA missing the velvet domain and VeA truncation strains (shown in A) after 18 h vegetative growth (hyphae) from an illuminated liquid culture as well as from cultures grown on tilted agar slides with illumination (conidiophore) for promoting asexual or darkness (cleistothecia) for sexual development. The nucleus is visualized with RFP-H2A (size bars: 10 µm).

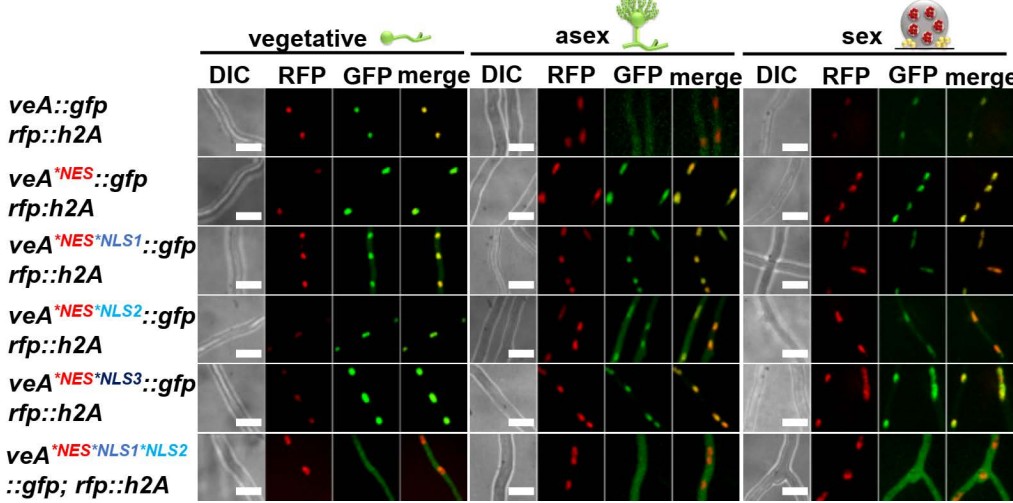

**Fig 3. Interplay between NES and NLS1 for combined action on VeA cellular localization.** Fluorescence images of strains with VeA-GFP and VeA-GFP with amino acid substitutions in NES or in NES/NLS motif combinations were compared after 18 h of vegetative growth and on agar slants after either 18 h in light (conidiophore) to promote asexual or in darkness (cleistothecia) to promote sexual development. Nuclei are visualized with RFP-H2A. A defective NES motif leads to an accumulated VeA-GFP signal in the nucleus. Combinations of defective NES and NLS reduce nuclear VeA accumulation, especially if NLS1 is dysfunctional. NES, NLS1 and NLS2 amino acid substitution strain shows less cytoplasmic signals (size bars: 10 µm).

nuclear accumulation in light of approximately 30% compared to wild type (S1 Fig). During darkness, the nuclear import of VeA remains possible without functionality of the three NLS, indicating that there is another unknown component involved in the cellular transport of VeA. However, any of the three NLS is required for the nuclear exclusion of VeA during illumination to ensure correct asexual development. Normally, the NLS of a protein enables its import into the nucleus. In this case, however, VeA NLS might have a potential contribution in the predominantly cytoplasmic localization and nuclear exclusion during illumination for supporting the asexual cycle of VeA.

Different truncations of VeA C-terminally fused to GFP were constructed, in addition to the *veA::gfp* fusion genes with point mutated codons for the signal sequences (Figs 1A and 2A). VeA$^{46-573}$-GFP missing the NLS1 as well as VeA$^{\Delta velvet}$-GFP lacking both NLS1 and NES resulted in nuclear signals during light (Fig 2C). Moreover, localization of VeA$^{1-225}$-GFP missing the NLS2 and NLS3 motifs as well as VeA$^{46-225}$-GFP additionally without the NLS1 motif resemble VeA variants with an increased nuclear accumulation of approximately 40% (S1 Fig). These findings suggest that there might be even another not yet identified NLS for the nuclear import of VeA. Moreover, the VeA NLS sequences do not only provide nuclear import, but also contribute to nuclear exclusion during illumination, which favors asexual and reduces sexual fungal development.

### The VeA NES is required to exclude the protein from the nucleus in light when asexual development is promoted

The C-terminal part of the VeA velvet domain contains a putative NES (Fig 1). The function of the VeA NES motif was assessed by codon exchanges and additional combinations with the amino acid substitutions of the NLS elements described before (Fig 1A). Fluorescence microscopy revealed that VeA wild type protein is mainly localized in the nucleus during vegetative growth (Figs 3 and S2). Surface growth during illumination (inducing asexual development) mostly excludes VeA from the nucleus. In contrast, growth in darkness (inducing sexual development) results in increased nuclear localization of VeA (Figs 3 and S2).

The VeA$^{*NES}$ resembles a similar protein distribution pattern like wild type during both vegetative and sexual growth conditions (Figs 3 and S2). However, the nuclear signal of VeA$^{*NES}$ increases more than 30% compared to intact VeA-GFP during asexual inducing conditions (S2 Fig). This suggests that the predicted NES indeed has a role in nuclear export during asexual development. These results corroborate that VeA is transported into nuclei during vegetative growth and illumination and the initiation of the asexual program subsequently results into the export of VeA. The export is reduced during sexual conditions. The cellular distribution of VeA during sexual development was unaffected in regard of the single NES mutation or in combination with any single defective NLS (VeA$^{*NES*NLS1}$, VeA$^{*NES*NLS2}$, VeA$^{*NES*NLS3}$) (Figs 3 and S2). The nuclear VeA fraction was reduced during vegetative growth and darkness, showing an almost exclusively cytoplasmic signal of VeA$^{*NES*NLS1*NLS2}$ (Figs 3 and S2). This suggests that dynamic VeA import and export processes are active during vegetative fungal growth. The results support a linked contribution of NLS1 and NLS2 in VeA nuclear import during vegetative growth. However, this leaves the contribution of NLS3 to nuclear import unclear. The nuclear VeA entry control is highly complex and might even require additional not yet identified sequences. These data support a convoluted interplay between NLS and NES signals resulting in the decision to inhibit or to favor VeA nuclear export with the complex outcome to promote either more sexual or more asexual development (Table 1).

## The combination of three VeA NLS motifs is required for the production of sufficient asexual spores and for cleistothecia maturation with the appropriate secondary metabolism

The microscopic analysis of the VeA-GFP variants revealed that a complex interplay between the NLS1-3 import and the NES export signals determines whether *A. nidulans* VeA remains in the nucleus or whether the developmental regulator

**Table 1. Summary of GFP-signal quantification from VeA variant localizations in the nucleus and cytoplasm during light and dark conditions.**

| Impaired Signal | *NLS1 | | *NLS2 | | *NLS3 | | *NES | |
|---|---|---|---|---|---|---|---|---|
| Condition | Light | Dark | Light | Dark | Light | Dark | Light | Dark |
| Nucleus | + | o | | | | | | |
| Cytoplasm | − | o | | | | | | |
| Nucleus | | | + | o | | | | |
| Cytoplasm | | | − | o | | | | |
| Nucleus | | | | | + | o | | |
| Cytoplasm | | | | | − | o | | |
| Nucleus | | | | | | | + | o |
| Cytoplasm | | | | | | | − | o |
| Nucleus | o | o | o | o | | | | |
| Cytoplasm | o | o | o | o | | | | |
| Nucleus | + | o | + | o | + | o | | |
| Cytoplasm | − | o | − | o | − | o | | |
| Nucleus | + | o | | | | | + | o |
| Cytoplasm | − | o | | | | | − | o |
| Nucleus | | | + | o | | | + | o |
| Cytoplasm | | | − | o | | | − | o |
| Nucleus | | | | | + | o | + | o |
| Cytoplasm | | | | | − | o | − | o |
| Nucleus | o | − | o | − | | | o | − |
| Cytoplasm | o | + | o | + | | | o | + |

The grey squares symbolize the single, double, or triple impaired NES/NLS signals. The circle means a protein distribution similar to wild type. The plus indicates an increased protein signal present, whereas the minus indicates a reduction with respect to VeA-GFP (based on S1 and S2 Figs).

is exported into the cytoplasm. It was examined whether these findings represent a molecular basis for the regulation of development, as VeA inside the nucleus supports sexual reproduction, whereas localization of VeA primarily outside of the nucleus promotes asexual development [15].

The colony phenotype of the *A. nidulans veA* deletion strain is significantly different to wild type with defects in the formation of cleistothecia [15] (Fig 4A). Additionally, the *veA* deletion strain shows a significantly reduced growth of about 30% compared to wild type as well as to a control strain with functional *veA::gfp*. The *veA::gfp* complementation strain shows growth similar to wild type (Fig 4B). An alteration of color, which is visible at the bottom of the colony reflects impaired control of secondary metabolite production (Fig 4A) [15]. The red-brownish color of the *veA* deletion strain might derive from the production of F9775A/B (II) derivatives (Fig 4C) as described for other deletion strains [7,21]. Wild type and the *veA::gfp* complementation strain had similar secondary metabolite profiles during development with following exceptions: sterigmatocystin (V), emericellin (VI), shamixanthone (VII), and epishamixanthone (VIII) were reduced during fungal development in the *veA::gfp* complementation strain, suggesting a possible influence by the *gfp*-fusion (Fig 4C).

The *veA*[*NLS1] and *veA*[*NLS2] variants resemble the wild type in fungal development (Fig 4A and 4B). The *veA*[*NLS3] strain shows a whitish color at the bottom of the colony, which is different to the colorization of the wild type and *veA* deletion strain (Fig 4A). This indicates an alteration in secondary metabolism in the strain with *veA*[*NLS3] variant. Similarly, secondary metabolism of the *veA*[*NLS1*NLS2] strain seems different compared to wild type visible by the light brownish colony color of the mutant strain (Fig 4A). Both strains produce smaller cleistothecia than wild type, suggesting a contribution in fruiting body maturation (Fig 4A). Combined amino acid substitutions in NLS1, NL2 and NLS3 (*veA*[*NLS1*NLS2*NLS3]) resulted in a similar phenotype as the *veA* deletion strain with altered color of the colony and a defect in the production of sexual fruiting bodies (Fig 4A). Xanthones are produced in Hülle cells which protect and therefore support the development of cleistothecia [8,9]. During sexual conditions, xanthone derivatives emericellin (VI), shamixanthone (VII), and epishamixanthone (VIII) were neither identified in the extracts of the *veA* deletion strain nor in those of the NLS1/2 and NLS1/2/3 variants (Fig 4C). This correlates with the phenotype of the NLS double amino acid substitution strain, which shows a delay in cleistothecia maturation, and the defect of cleistothecia production in the NLS triple amino acid substitution strain. Therefore, all three VeA NLS sequences participate in the regulation of fungal secondary metabolism and cleistothecia formation (Fig 4A and 4C).

The colony size of each strain with amino acid substitutions in NLS motifs shows a significant growth reduction of 10–20% compared to the *veA::gfp* strain. VeA is important for asexual development, because a respective deletion strain produces only 40% of the conidiospores compared to *veA::gfp* after five days of surface growth in light. NLS3 seems to be most important because its inactivation results in a strain with similar reduction in spore production as the *veA* deletion strain, whereas changes in the other NLS did not significantly change sporulation (Fig 4B). This shows that control of VeA cellular localization by the three VeA NLS signals and especially NLS3 is important for accurate spore production during asexual development. Dehydroaustinol is required for sporulation of *A. nidulans* [22]. Austinol (III) and dehydroaustinol (IV) amounts were reduced in the *veA*[*NLS3] strain compared to wild type during fungal development (Fig 4C). This correlates with the observed decreased spore amount of the NLS3 amino acid substitution strain and corresponds to the observations with the *veA* deletion strain (Fig 4A and 4C).

Taken together, the combined three NLS motifs of VeA are required in conidiospore development and cleistothecia maturation in combination with the corresponding secondary metabolism. They seem mandatory for the formation of the required amount of austinol, dehydroaustinol, or xanthone derivatives.

## VeA nuclear export is required to allow coordinated development and secondary metabolism in *A. nidulans*

Fluorescence microscopy studies have shown that the NES motif is required to exclude VeA from the nucleus in light when asexual development should be promoted. Fungal growth and developmental phenotypes caused by non-functional NES and NLS signals were compared to corresponding wild type and *veA* deletion strains. The *veA*[*NES] variant as well as the different *veA*[*NES*NLS] variants resemble the *veA* deletion phenotype both in asexual and sexual development (Fig 5A).

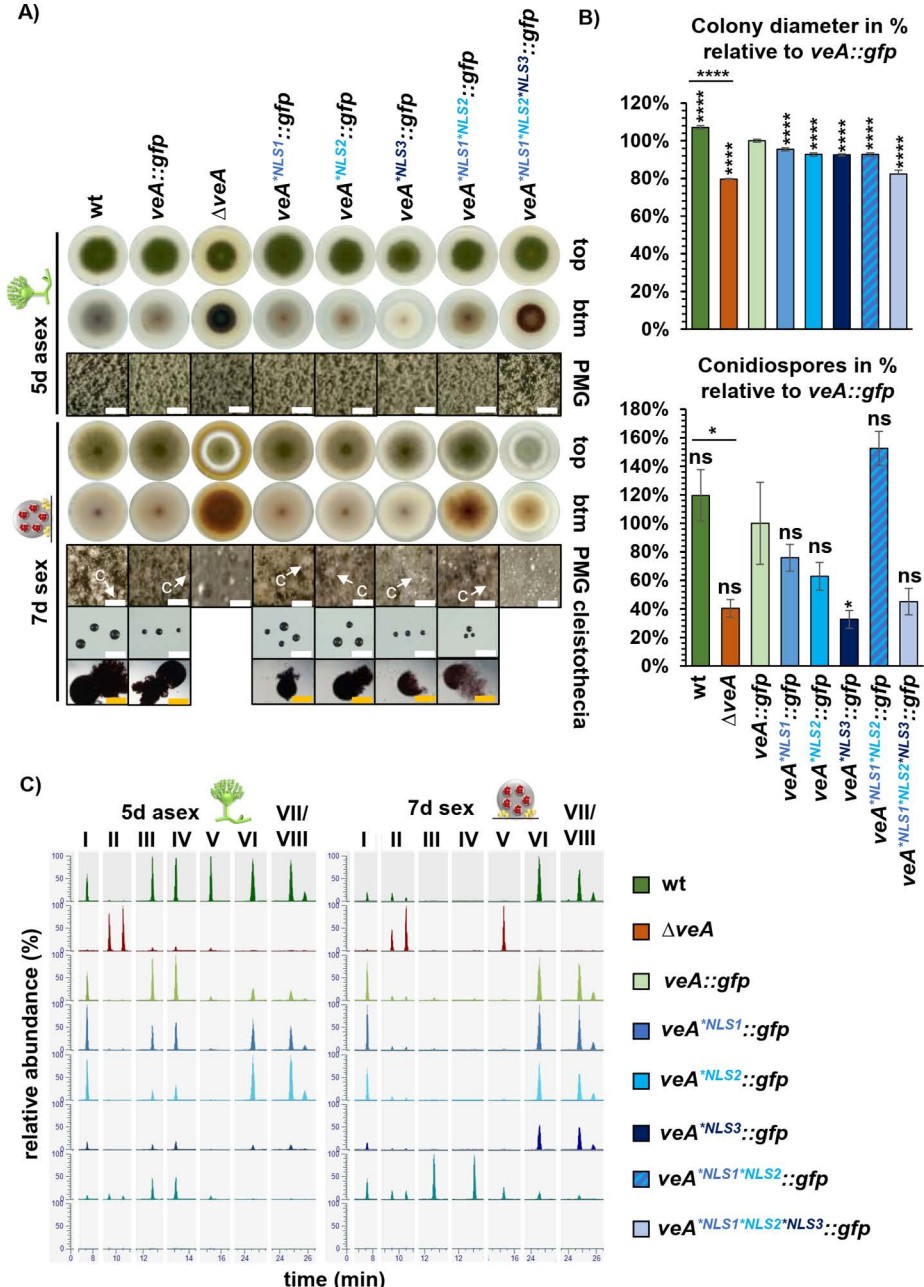

**Fig 4. All three NLS of VeA are required for wild type like development in *A. nidulans*.** (A) Asexual development of *veA* deletion (Δ*veA*) and complementation (*veA::gfp*) strains as well as strains with amino acid substitutions in the NLS motifs (*veA*^*NLS1*^::*gfp*, *veA*^*NLS2*^::*gfp*, *veA*^*NLS3*^::*gfp*, *veA*^*NLS1*NLS2*^::*gfp*, *veA*^*NLS1*NLS2*NLS3*^::*gfp*) and the wild type were analyzed after 5 days of incubation at 37°C in light. Plates were incubated in the dark and with limited oxygen for 7 days at 37°C for sexual development. (PMG: photomicrographs, c: cleistothecia, size bar: 200 μm (white), cleistothecia 50 μm (orange)). (B) Measurement of colony diameters and quantification of spores from wild type, *veA* deletion and mutant strains. The diagram summarizes quantification results from three biological replicates (3 technical replicates each). Error bars represent standard error of the mean, and the significances were compared to *veA::gfp* complementation strain or between strains with connecting lines (*p* > 0.05: ns, *p* ≤ 0.05: *, *p* ≤ 0.0001: ****). (C) Ethyl acetate extracts were obtained from the indicated strains that were cultivated as described in A. Samples were measured by LC-MS. Extracted ion chromatograms (EIC, [M-H]⁻) of the most prominent identified masses show differences in peak intensities between the analyzed strains. Identified metabolites are chichorine (I), F-9775A/B (II), Austinol (III), Dehydroaustinol (IV), Sterigmatocystin (V), Emericellin (VI), Shamixanthone (VII), and Epishamixanthone (VIII). Detailed information about identified metabolites is given in S1 Table. A tolerated mass deviation of 0.0010 was used to visualize the metabolites.

A similar growth reduction of the mutant strains as well as *veA* deletion could be observed compared to *veA::gfp* complementation strain (Fig 5B). Sporulation is not significantly different between all tested strains (Fig 5B). All these strains showed the dark brown color at the bottom of the colony as indicator of altered secondary metabolisms in combination with the inability to form cleistothecia during sexual development. In contrast, wild type and *veA::gfp* complementation strain completed the sexual cycle and produced cleistothecia.

Wild type and *veA::gfp* complementation strains were additionally compared with truncated VeA versions during fungal development (S3 Fig). The strains with a truncated VeA missing either one or more of the NES and NLS motifs resemble the *veA* deletion phenotype both in asexual and sexual development underlining that the motifs as well as the velvet domain are required for the functionality of the VeA protein (S3 Fig). Additionally, the secondary metabolite composition of the NES variant strains resembled that of the *veA* deletion strain (Fig 5C). The only exception was found for the $veA^{*NES}$, $veA^{*NES*NLS2}$, and $veA^{*NES*NLS3}$ strains, which produce sterigmatocystin (V) during sexual development (Fig 5C and S1 Table). The amount is comparable to wild type in $veA^{*NES*NLS2}$ and even increased in the $veA^{*NES*NLS3}$ strain. This indicates that these NES and NLS motifs, which regulate the cellular localization of VeA, provide an important function in regulating secondary metabolism. Furthermore, it supports that the distribution of VeA in the nucleus or in the cytoplasm is directly coupled to the coordination of development with the corresponding secondary metabolism and thereby suggesting a connection to developmental regulation.

## *A. nidulans* development relies on coordinated nuclear transport of VeA which requires correct protein levels

It was examined whether similarities in phenotype between the *veA* deletion strain and all strains with amino acid substitutions in the NES are due to reduced or instable VeA protein amounts resulting from introduced amino acid substitutions. Gene expression of *veA::gfp* was compared by qRT-PCR with *veA* variants during vegetative growth to examine whether mutations of codons of the *veA* localization signal influence the transcription level. Gene expression is increased in the strain expressing *veA::gfp* with the NES single mutation as well as the double mutations with one of the NLS compared to wild type *veA* during vegetative growth (Fig 6A). Only $veA^{*NES*NLS1*NLS2}::gfp$ showed a similar expression level to the *veA::gfp* strain (Fig 6A). Increased transcription presumably also results in increased protein amounts during vegetative growth, which might be a response of the cell to the non-functional VeA. More protein is produced to possibly ensure the fulfillment of VeA's tasks.

Therefore, the protein amount of respective strains was analyzed during vegetative growth as well as fungal development. Most strains depict an increased protein amount. The amount of protein is significantly increased in VeA$^{*NES*NLS2}$-GFP (Figs 6 and S4). This is different for VeA$^{*NLS1}$-GFP and VeA$^{*NES*NLS1*NLS2}$-GFP. The former has an increased transcription level, but the protein amount is similar to wild type VeA-GFP (Figs 6 and S4). $veA^{*NES*NLS1*NLS2}::gfp$ has no significant change in the transcription level. The amount of the encoded protein during vegetative growth is also similar to wild type VeA-GFP. The protein abundance of wild type VeA-GFP significantly decreases after 6–12 hours of sexual development and after 18 hours of asexual development showing that VeA-GFP is more stable under asexual growth conditions (Figs 6 and S4). In contrast, the amino acid substitutions in the NES and NLS motifs lead to a more stable protein under sexual development compared to wild type VeA-GFP (Fig 6). An exception is VeA$^{*NES*NLS1}$-GFP which resembles the protein level of VeA-GFP (Fig 6). However, the amino acid substitutions in the NES and NLS motifs have the opposite effect during asexual development. They lead to a less stable VeA-GFP, whereas under this condition VeA$^{*NLS2}$-GFP is an exception, resembling the VeA-GFP protein level (S4 Fig).

Taken together, all tested NES and NLS motifs are required to provide accurate VeA protein levels for fungal development. However, defects in NLS1 and NLS2 are tolerated by the cell concerning protein translation during asexual and sexual development, respectively, because strains with mutations in these motifs show VeA-GFP wild type-like protein levels under the respective conditions. In summary, the NES and NLS motifs of VeA are not only required for correct localization in nucleus or cytoplasm during vegetative growth or fungal development, but also to maintain sufficient VeA protein levels.

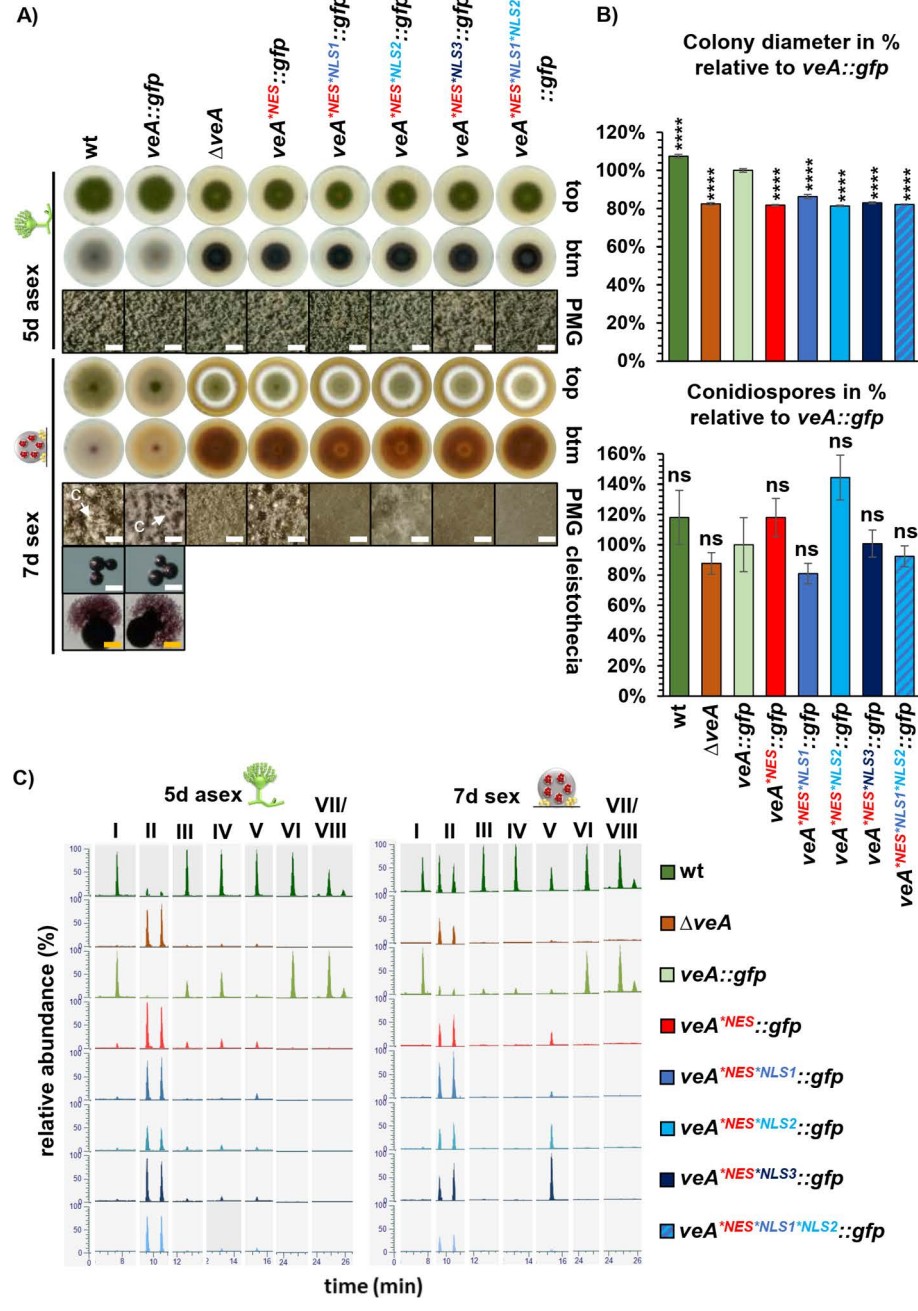

**Fig 5. Nuclear export of VeA is required for *A. nidulans* development and coordinated secondary metabolism.** (A) Wild type, the *veA* deletion strain as well as strains with amino acid substitutions in NES and NLS motifs were analyzed five days after point inoculation on solid MM and incubation at 37°C in light to induce asexual development. For sexual development, plates were incubated in the dark and with limited oxygen for 7 days at 37°C. (PMG: photomicrographs, c: cleistothecia, size bar: 200 μm (white), cleistothecia 50 μm (orange)). (B) Measurement of colony diameters and quantification of spore amounts from wild type, *veA* deletion and mutant strains. Quantification results from three biological replicates (3 technical replicates each). Error bars represent standard error of the mean, and the significances were compared to *veA::gfp* complementation strain (*p* > 0.05: ns, *p* ≤ 0.0001: ****). (C) Ethyl acetate extracts were obtained from the indicated strains that were cultivated as described in A. Samples were measured by LC-MS. Extracted ion chromatograms (EIC, [M-H]⁻) of the most prominent identified masses show differences in peak intensities between the analyzed strains. Identified metabolites are chichorine (I), F-9775A/B (II), Austinol (III), Dehydroaustinol (IV), Sterigmatocystin (V), Shamixanthone (VII), and Epishamixanthone (VIII). Detailed information about identified metabolites is given in S1 Table. A tolerated mass deviation of 0.0010 was used to visualize the metabolites.

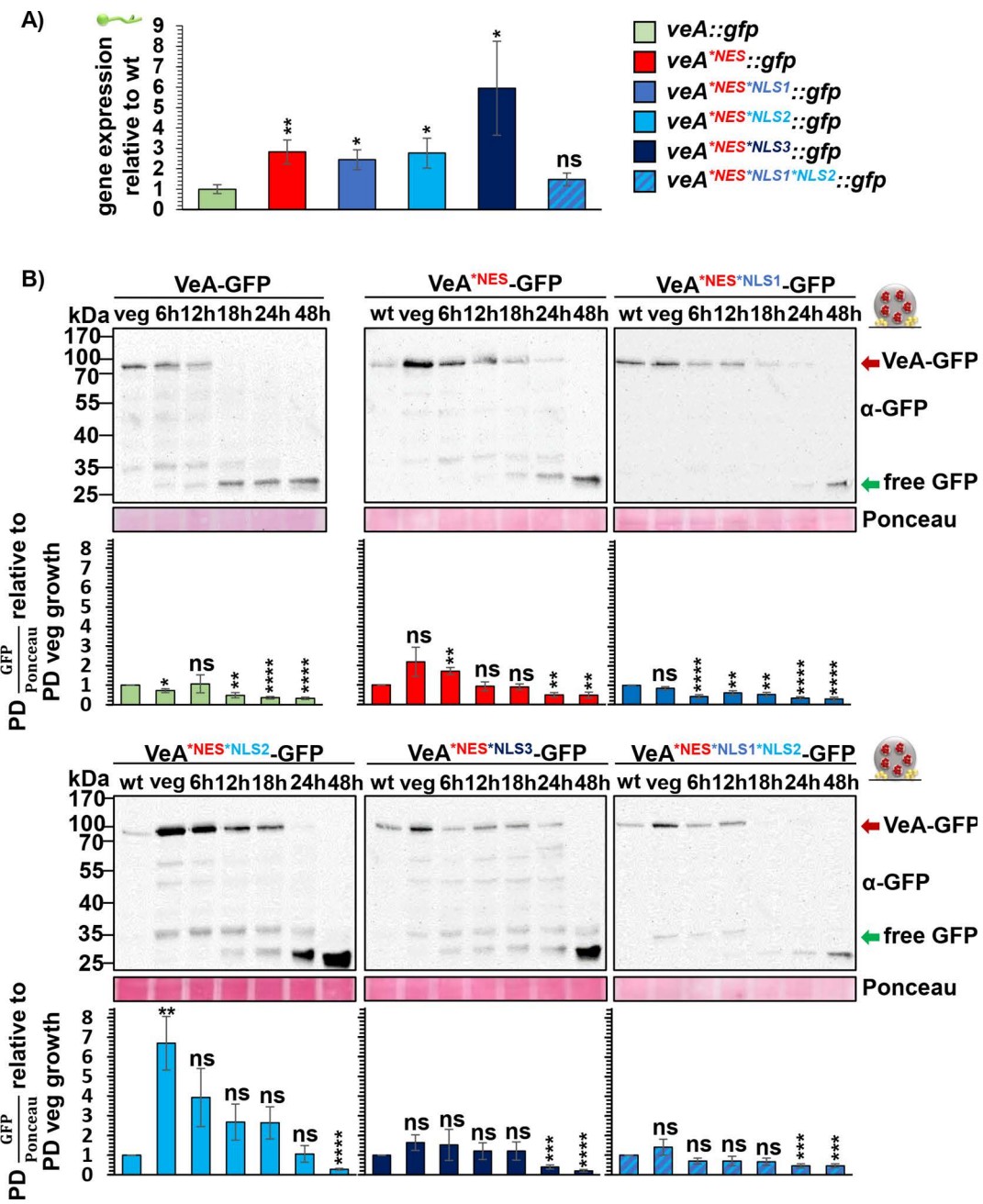

**Fig 6. Transcription and VeA protein levels depend on VeAs NES/NLS during vegetative growth or sexual development.** (A) qRT-PCR was performed to analyze gene expression of *veA::gfp* and mutant strains. Fungal strains were grown for 20 h in liquid cultures for vegetative growth. RNA was isolated from these samples and transcribed to cDNA. qRT-PCRs were performed with *h2A* and *gpdA* as reference genes. Expression levels of the mutants were quantified relative to *veA::gfp* expression. Quantification results from three biological replicates (3 technical replicates each). (B) Western experiments of fungal protein extracts from VeA-GFP and VeA NES and NES/NLS mutant strains were performed. All strains were grown for 20 h at 37°C in liquid minimal medium for vegetative growth. Mycelia were shifted onto 30 ml minimal medium agar plates for 6 to 48 h at 37°C in dark for sexual development. Samples were collected at identical time points and protein crude extracts were prepared for western experiments with α-GFP antibody. Signal quantification was performed using BioID software, signals of VeA-GFP (red arrow) were normalized to Ponceau S staining. Wild type (wt) VeA protein amount at the 20 h vegetative time point was set to 1 and used as reference for protein amounts of following time points of wild type VeA or VeA with amino acid substitutions in NES and NLS motifs. Quantification results from three biological replicates (2 technical replicates each). Error bars for western experiments and qRT-PCRs represent standard error of the mean ($p > 0.05$: ns, $p \leq 0.05$: *, $p \leq 0.01$: **, $p \leq 0.001$: ***, $p \leq 0.0001$: ****).

Cleistothecia formation in *A. nidulans* sexual development is the essential function of the NES motif, which is connected to the formation of Hülle cells providing nursing functions and a protective secondary metabolism [8,9].

## Discussion

The VeA velvet domain protein is a key regulator of fungal development. VeA in complex with other (velvet) transcription factors induces asexual as well as sexual spore formation in coordination with developmental program linked secondary metabolite biosynthesis in *A. nidulans* [9,10,15]. Gene expression requires a tight control of VeA being present or absent from the nucleus at the right time during fungal development. Nuclear transport of VeA is light dependent [23]. VeA is found in the nucleus in the dark (sexual conditions) and can be observed in the cytoplasm in light (asexual conditions) [15]. VeA is transported into the nucleus by interacting with the importin KapA as a heterodimer with VelB [13].

This study revealed a complex shuttle control mechanism for VeA localization either in the nucleus or in cytoplasm during fungal differentiation. VeA is already imported into the nucleus during early fungal growth and remains there during the vegetative phase (Fig 7). Three nuclear localization sequences as well as an export signal are required for translocation of the VeA protein during differentiation from cytoplasm to nucleus and back. The different VeA locations, which are provided by this shuttling process promote the transition from *A. nidulans* vegetative growth either to the asexual (cytoplasmic VeA) or to the sexual (nuclear VeA) developmental program and its corresponding specific secondary metabolisms.

Asexual development requires that VeA, which had been imported during vegetative growth is exported and excluded from the nucleus (Fig 7). Nuclear mislocalization of VeA by a defective export signal causes a block in cleistothecia production and altered secondary metabolism during illumination, similar to a strain carrying a deletion in the *veA* gene. An impaired NLS3 motif resulted in decreased production of austinol and dehydroaustinol. Dehydroaustinol is necessary for spore production in *A. nidulans* [22]. Consistently, strains with impaired NLS3 have a significantly reduced spore production. This supports that the combination of an intact NLS3 import and the NES export signal is prerequisite for the potential for asexual development. The conidia production in the other NLS/NES variant is not altered by the nuclear stay of VeA during asexual development. VeA could have through NLS3 an impact on the localization of other yet unknown interaction partners through co-transport as dimers. These interaction partners might play a decisive role in controlling fungal development and could be addressed in future studies. It is likely that these VeA interaction partners are required for transport processes either to the periphery or into or out of the nucleus. This supports that the cellular localization dynamics and its machinery are even more complicated than already assumed for controlling fungal development.

It is yet elusive, whether the VeA exclusion process during asexual development might include VeA posttranslational modifications (PTMs) such as phosphorylation or acetylation [24] or NLS interactions with components of the nuclear pore complex (NPC). Monoubiquitinaton next to an NLS signal can also contribute to nuclear exclusion due to disrupted importin interaction [25]. The effect of the altered NLS motifs on potential PTMs will be an interesting aspect for further research.

Defects in VeA NLS motifs and false localization had also effects on sexual development as delay or even failure in cleistothecia maturation. This correlates with reduction or even complete impairment in xanthone derivative (emericellin, shamixanthone, epishamixanthone) production. Xanthones are produced in nursing and protective Hülle cells and defend ascospores from predators [8,9]. They support the development of cleistothecia. The respective gene expression might not be accurately controlled by a non-functional VeA, either alone or in combination with other velvet proteins [25]. This results in delay of cleistothecia maturation or even a complete loss as observable in the *veA* mutant strains. VeA NLS motifs are mandatory to regulate xanthone biosynthetic genes from the *mdp* and *aus* secondary metabolite gene clusters and thereby control cleistothecia maturation and spore production.

Nuclear export is a critical step in regulating cellular functions [26]. A defective NES motif leads to an accumulation of VeA in the nucleus and a fungal developmental phenotype with impaired secondary metabolism, which resembles a corresponding deletion strain where VeA is absent. The correct nuclear export of VeA is therefore mandatory for fungal development and

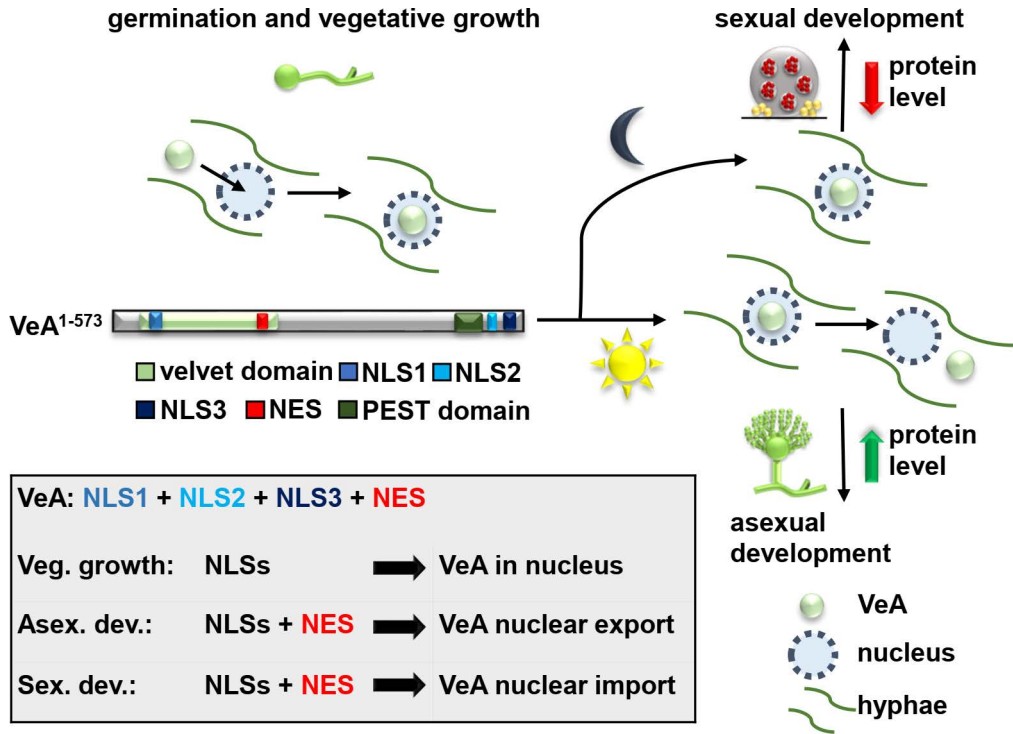

**Fig 7. VeA shuttles between cytoplasm and nucleus to allow *A. nidulans* fungal development.** VeA is channeled into the nucleus during germination and vegetative growth. External signals as darkness or decreased oxygen levels lead to VeA-induced sexual development forming, cleistothecia as hibernating structures. Light or increased oxygen levels lead to nuclear export of VeA to the cytoplasm for accurate asexual development. The nuclear exchange of VeA is connected with a stability control mechanism of the velvet protein for both developmental programs.

its link to the appropriate secondary metabolism. The retention time of proteins like VeA with an NES in the nucleus needs to be timely regulated [27]. Amino acid changes in the VeA NES motif might impair interactions to other regulatory proteins or hinder nuclear transport of other proteins. The interaction of VeA with VelB and therefore the heterodimer formation might be disturbed by the defective motifs of VeA, since both proteins together regulate the initiation of sexual development [15,28]. Also the formation of the velvet complex with LaeA to regulate secondary metabolism could be hindered therefore leading to the phenotypical effect of the VeA variants [15]. Another possibility would be that VeA interaction partners are required to be backpacked out of the nucleus by VeA. Moreover, VeA needs to be exported from the nucleus to allow the expression of genes required for asexual development. Additionally, a constant import and export of VeA is required to control gene expression required for sexual development, like *steA* (sterile12-like) or *esdC* (early sexual development). SteA controls cleistothecia development and ascospore production [29], whereas EsdC acts also on repression of conidiation [30].

The combination of NLS1, NLS2 and NES motifs is required for the nuclear import of VeA during vegetative growth and subsequent later sexual development (Fig 7). The bipartite NLS3 seems to have a specific function, because it supports spore production during asexual development and might cooperate with the NES. Defective NES/NLS VeA variants are predominantly cytoplasmic and might no more be able to interact with the KapA importin or other participating interaction partners for nuclear transport. The accessibility to the NLS or NES motifs could have been changed by one or more of the amino acid substitutions. These defects could lead to a conformational change of the VeA protein and thus its localization by enabling or disabling interactions with proteins like KapA or the LaeA-like methyltransferase LlmF, which impacts the nuclear/cytoplasmic ration of VeA [31]. VeA stays in the nucleus after reaching the competence state to induce sexual development or is then excluded from the nucleus back to cytoplasm for an accurate asexual development.

False localization of VeA during development and the resulting phenotypical defects show how important correct localization is for the function and stability of transcription factors [32]. The PEST domain of VeA is presumably needed for a rapid protein turnover of VeA [33]. Proteins with such a PEST domain have a specific half-life and the sequence directs the protein to degradation. Cytoplasmic localization seems to prevent VeA from degradation and therefore stabilizes VeA in the cytoplasm (Fig 7). Additionally, the *veA* variants result in an increased *veA* transcription level during vegetative growth leading to a higher protein amount at the same condition (Fig 6). This shows a response of *A. nidulans* on the non-functional VeA to retain somehow the function of VeA during fungal development, suggesting a feedback-regulation. Moreover, the amino acid substitution in the VeA motifs could have additionally changed potential ubiquitination sites. VeA has three predicted ubiquitination sites: one in the NLS1 motif, the second at the end of the velvet domain, and a third one in the NLS3 motif [33]. The first and second ubiquitination sites were not affected by the amino acid substitutions. Ubiquitin actually binds to lysine residues. The lysine residue was changed in the NLS3 motif. The altered PTM could have changed the time point or the way of modification of VeA, possibly leading to a more stable VeA. In addition to the ubiquitination sites, a structural change could alter the protein abundance. A structural alteration could affect the PEST domain localized in the N-terminus of VeA, where also the NLS3 is found.

VeA is a highly conserved fungal regulator of development and secondary metabolism [11,15]. This includes the opportunistic pathogen *Aspergillus fumigatus*, causing invasive fungal infections in immunocompromised patients [34,35] or *Aspergillus flavus*, which contaminates 60–80% of the worldwide crops post-harvest, causing aflatoxicosis in both humans and animals after consumption of spoiled food [36]. Additionally, the VeA counterpart of the plant pathogen *Verticillum dahliae* is required for plant root colonization and conidia formation for distribution in the vascular plant system [37]. An alignment of the VeA motifs by Clustal Omega protein sequence alignment [38] shows that NLS1 and NLS2 are conserved in *Aspergillus* species, *Verticillium dahliae* and *Neurospora crassa* (S5A and S5C Fig) [15,37]. The NES motif seems to be conserved in these *Aspergillus* species since they share the amino acid sequence. The NES is not present in either *N. crassa* or *V. dahliae* (S5B Fig). The alignment of the NLS3 shows that the sequence might be conserved in Aspergilli (S5D Fig). Another interesting aspect for future studies would be to analyze the effect of the different motifs on, e.g., the pathogenicity of other *Aspergillus* species.

In conclusion, we demonstrated that the conserved fungal developmental velvet domain regulator VeA is localized in the nucleus during vegetative growth in early fungal life cycle (Fig 7). It remains nuclear during sexual fruiting body resting structure formation in darkness, which might be the default differentiation program of *A. nidulans*. VeA becomes exported in light to enable asexual spore formation for distribution in the air. This regulation of VeA localization allows the fungus to react to changing environmental conditions and to adapt its development accordingly. It might be interesting to explore whether and how often the fungus can still adjust differentiation to quickly changing environmental conditions. The fungal velvet domain shows the same structural fold as NF-κB and its relatives including conserved DNA binding amino acid residues [14]. The NF-κB family represents important transcription factors for the immune cell response of mammalian cells [39]. It will be interesting to see whether the nuclear localization control mechanisms observed for fungal VeA are also reflected in the corresponding human proteins. The results corroborate the importance of a tight control of the nuclear import as well as export for a central transcription regulator controlling fungal distribution through the air (by asexual spores), overwintering and resting structure formation in the soil (cleistothecia) with sexual spores which are spread through water combined with the corresponding protective formation of secondary metabolites as chemical fungal language.

## Methods

### Databases and identification of potential signal motifs

Sequences for *Aspergillus* strain construction were obtained from AspGD [40], FungiDB [41] and UniProt [42]. NLS prediction for VeA was conducted with cNLS mapper [18]. Additionally, the LocNES mapper tool [17] was used for the prediction of the NES in the velvet domain. For both predictions the standard settings of each tool were used.

## Strains and growth conditions

The *E. coli* strain DH5α was cultivated in in liquid lysogeny broth (LB) medium [43]. For the solid medium 2% agar was added. Selection was performed with 100 mg/mL Ampicillin.

*A. nidulans* was cultivated in liquid or solid minimal medium (MM) [44]. The *A. nidulans* strain AGB552 [45] was used as a parental strain. Therefore, 0.1% (v/v) 1 M para-amino-bezeoacid (*paba*) was added to liquid and solid MM. For the recyclable marker cassette pME4319 containing the *ble* gene resistance to phleomycin was used [46]. For the selection of strains after transformation, 0.1% (v/v) 1 M phleomycin was added to the medium. Marker cassette recycling was carried out on medium containing 0.5% (w/v) glucose and 0.5% (w/v) xylose as carbon source. Spores on agar plates were incubated at 37°C in light to induce asexual development or in darkness for sexual development by sealing the plates with parafilm (Bemis). For vegetative growth, fungi were incubated in liquid MM in baffled flasks for 20 hours at 37°C with agitation. For time point experiments, mycelia were poured through filters and then shifted on 30 ml MM plates.

## Plasmid and strain construction

The primers and plasmids were designed using the DNASTAR Lasergene software package (version 12.1.0) from DNASTAR Inc. The *E. coli* strain DH5α was used for plasmid construction with seamless cloning (Invitrogen GeneArt Seamless Cloning and Assembly Kit, Thermo Fisher Scientific) or Quick Change site-directed mutagenesis (Agilent Technologies). All oligonucleotides used for the construction of plasmids are given in S3 Table. The quick-change protocol [47] was performed to exchange codons within all potential nuclear import or export sequences for the indicated amino acid residues (Fig 1A) to assess their molecular cellular function in VeA localization. Codons for basic lysine and arginine residues of each NLS as well as for the hydrophobic leucine residue of the NES were changed as indicated to non-essential, non-polar alanine residues [48]. The transformation with the *E. coli* strain DH5α was performed as previously described [49]. Plasmids were isolated via QIAprep Spin Miniprep Kit (Qiagen) and sequenced to confirm presence of the desired mutation (Microsynth Seqlab AG).

*A. nidulans* transformations were performed as described elsewhere [50]. Strains were verified by Southern hybridization [51]. Plasmids used for the transformations are listed in S2 Table, and resulting strains used or prepared in this study are listed in S3 Table.

**Plasmid and strain construction of VeA$^{1-224}$.** Cloning of C-terminal Strep II-tagged VeA1-224 was previously described [12].

**Plasmid and strain construction of the VeA$^{1-225}$-GFP truncation.** For the construction of the VeA$^{1-225}$ C-terminally fused to GFP, the 3' flanking region of *veA* was amplified with oAS40/oAS41 (1663 bp). *gfp* was amplified from gDNA of AGB1351 with oAS38/oAS39 (732 bp). *veA$^{1-225}$* and its 5'flanking region was amplified from gDNA of AGB1351 with oAS35/oAS37 (2448 bp). The 3' flanking fragment was ligated by seamless cloning into a site of *Pml*I linearized vector pME4319 and transformed into DH5α. For the second cloning step, the vector with the 3'flanking region of *veA* (further called 3'*veA*) was linearized with *Swa*I and then ligated with the 5'flanking region with *veA$^{1-225}$* by cloning, resulting in pME5533. The plasmid was cut with *Pme*I and the cassette (9741 bp) was transformed into AGB1337 resulting in AGB1666 after marker recycling.

**Plasmid and strain construction of the VeA$^{46-225}$-GFP truncation.** The C-terminal GFP tagged VeA$^{46-225}$ was constructed by amplification of 5' flanking region of *veA* by oAS35/oAS36 (1718 bp) and *veA$^{46-225}$* by oAS42/oAS37 (598 bp) from gDNA of AGB1351, as well as the *gfp* with oAS38/oAS39. The vector 3'veA was linearized by *Swa*I and the fragments ligated with seamless cloning resulting in pME5534. The plasmid was *Pme*I digested, and the cassette was transformed to AGB1337 resulting in AGB1667 after marker recycling.

**Plasmid and strain construction of VeA$^{*NES}$-GFP.** To create a point mutation in the NES in the velvet domain of VeA, 5'flanking region of *veA* and *veA* fused with *gfp* was amplified from AGB1351 gDNA with oAS35/oAS39 (4224 bp) and

ligated into the *Swa*I digested 3'*veA.* Then, primer oAS127/oAS128 were used to amplify the plasmid *veA::gfp* (13.761 kb) via Quick Change Protocol, creating point mutations L188A and L192A. The product was then transformed to DH5α. The resulting plasmid pME5544 was linearized with *Pme*I and transformed to AGB1337, creating AGB1677 after marker recycling.

**Plasmid and strain construction of VeA*NES*NLS2-GFP.** To create a point mutation in the putative monopartite NLS in the C-terminus of VeA additional to the NES point mutation, the Quick Change protocol was followed. Therefore, the total plasmid was amplified with oAS130/oAS129 (13.761 kb), creating point mutations K509A and R510A. The resulting plasmid pME5546 from the protocol was then *Pme*I digested and transformed into AGB1337, creating AGB1678 after marker recycling.

**Plasmid and strain construction of VeA*NES*NLS3-GFP.** Creating additional to the NES point mutation a point mutation in the putative bipartite NLS in the C-terminus of VeA, the plasmid pME5544 was used as a template. The 5' of *veA*, as well as amino acids 2442 bp to 4150 bp of *veA* was amplified with oAS35/oAS111 (3384 bp), whereas from amino acid 4137 bp of the C-terminus of *veA* that was fused to *gfp*, was amplified with oAS112/oAS39 (762 bp). Both fragments were fused into *Swa*I digested 3'*veA* by seamless cloning. Afterwards, the *Pme*I digested cassette pME5547 was transformed into AGB1337, resulting in AGB1679 after marker recycling.

**Plasmid and strain construction of VeA*NES*NLS1-GFP.** For the creation of a point mutation in the bipartite NLS in the velvet domain of VeA additional to the NES point mutation, the Quick Change protocol was followed. This was performed in two steps. The first step was to create the point mutation K28A and K29A by amplifying the total plasmid with oAS137/oAS138 (13.761 kb). The correct plasmid was then used as a template in a second step to amplify the plasmid with oAS135/oAS136 (13.761 kb). The resulting plasmid pME554 from the second step was digested with *Pme*I and transformed into AGB1337, resulting in AGB1680 after marker recycling.

**Plasmid and strain construction of VeA*NES*NLS1*NLS2-GFP.** For the construction of the VeA triple mutation of its NES, NLS1 and NLS2, the plasmid pME554 was used as a template to introduce the NLS2 point mutation via Quick Change protocol. Therefore, the same procedure was followed as for the plasmid pME5546. The resulted plasmid pME5549 was digested with *Pme*I and the cassette containing the gene for the desired fusion protein was transformed into AGB1337, resulting in AGB1681 after marker recycling.

**Plasmid and strain construction of VeA*NLS1*NLS2-GFP.** To construct a VeA*NLS1*NLS2 point mutation strain with a GFP tag, the plasmid pME5549 was used as a template to revoke the NES point mutation K509A and R510A with the Quick Change protocol. Therefore, the whole plasmid was amplified with oAS139/oAS140 (13.761 kb). Afterwards, the product was transformed to DH5α. The resulting plasmid pME5553 was linearized with *Pme*I and transformed into AGB1337, creating AGB1690 after marker recycling.

**Plasmid and strain construction of VeA*NLS2-GFP.** For the construction of a VeA*NLS2 point mutation strain with a GFP tag, the plasmid pME5546 was used as a template to revoke the NES point mutation K509A and R510A with the Quick Change protocol. Therefore, the whole plasmid was amplified with oAS139/oAS140 (13.761 kb). The product was then transformed to DH5α. The resulted plasmid pME5554 was linearized with *Pme*I and the desired fragment was transformed into AGB1337, creating AGB1691 after marker recycling.

**Plasmid and strain construction of VeA*NLS3-GFP.** For the construction of a VeA*NLS2 point mutation strain with a GFP tag, the plasmid pME5547 was used as a template to revoke the NES point mutation K509A and R510A with the Quick Change protocol. Therefore, the whole plasmid was amplified with oAS139/oAS140 (13.761 kb). The product was then transformed to DH5α. The resulted plasmid pME5555 was linearized with *Pme*I and the desired fragment transformed to AGB1337, creating AGB1692 after marker recycling.

**Plasmid and strain construction of VeA*NLS1*NLS2*NLS3-GFP.** For VeA*NLS1*NLS2*NLS3 -GFP, the plasmid pME5547 was used as a template to create the additional point mutation of NLS1 with the Quick Change protocol to create the preliminary plasmid pME5550. Therefore, the whole plasmid was amplified with oAS137/oAS138 (13.761 kb). The

resulting plasmid was used to create the precursor plasmid pME5551, where a point mutation in NLS2 was introduced by amplification with oAS129/oAS130. The precursor plasmid pME5551 were used as a template to revoke the NES point mutation K509A and R510A with the Quick Change protocol by amplification with oAS139/oAS140 (13.761 kb). Afterwards the protocol for Quick Change was followed and the product was then transformed to DH5α. The resulted plasmid pME5557 was linearized with *Pme*I and transformed into AGB1337, creating AGB1700 after marker recycling.

## Isolation of fungal genomic DNA and RNA and cDNA synthesis

gDNA was extracted as described before [52] whereas two changes in the protocol were conducted: *β*-mercaptoethanol was added to the gDNA extraction buffer as well as an additional step with 3 M sodium acetate. RNA was extracted by using the protocol of the NucleoSpin RNA Plant (Macherey-Nagel) according to manufacturer's instructions. cDNA was transcribed from the extracted RNA. The transcription was done with 2 µg RNA with the RevertAid Reverse Transcriptase Kit (Thermo Fisher Scientific), following the provided protocol.

## Spot test for phenotype analysis

For phenotypical characterization, strains were incubated at 37°C for 5 days to observe asexual development and for 7 days in the dark, sealed with parafilm to reduce the air fluctuation, to induce sexual development. Therefore, 4000 spores of respective strains were point inoculated on 30 ml MM plates. After 5 days, the size of the colonies was measured and for quantification of conidiospores were counted with a Thoma cell counting chamber (Thermo Fisher Scientific). After 7 days of sexual growth, the cleistothecia were examined with a binocular microscope SZX12-ILLB2-200 (Olympus Deutschland GmbH) and Axiolab light microscope (Zeiss).

## Protein Isolation and Western experiments

Protein extraction, SDS-PAGE, and Western experiments were conducted as described before [53]. The wild type VeA crude extract at the 20 h vegetative time point was loaded on each SDS-gel as reference for the followed quantification. The primary antibody was monoclonal α-GFP antibody (B-2, sc-9996) (Santa Cruz Biotechnology) and the secondary antibody α-mouse (horseradish peroxidase-coupled) (Jackson Immunoresearch Laboratories Inc.). The signals were visualized with the Fusion-SL7 chemiluminescence detection system (PeqLab Biotechnologie GmbH) and Fusion software (Vilber Lourmat Deutschland GmbH). The GFP-signals and Ponceau were analyzed with the Bio1D software (Version 15.08, Vilber Lourmat Deutschland GmbH). For normalization of the data, the Ponceau signal was used. The calculation of the significance was done with the homepage https://www.graphpad.com (accessed 2021–2023, GraphPad Software) using the unpaired t-test for two groups by using mean, SEM (standard error of the mean) and number of samples.

## Quantitative Real Time PCR

Transcription of *veA* was analyzed with genes encoding histone *h2A* and *gpdA* promotor served as housekeeping genes. Oligonucleotides used for qRT experiments are given in S4 Table. CFX Connect Real Time System cycler (Bio-Rad Laboratories Inc.) was used to analyze the transcription levels. Therefore, Mesa Green qPCR Master Mix Plus for SYBR Assay (Eurogenetec) and the corresponding standard protocol was used. Expression levels of *veA* was quantified with the ΔΔCT method [54] by using CFX Manager™ software version 3.1 (Bio-Rad Laboratories Inc.).

## Fluorescence microscopy

*A. nidulans* strains were inoculated in 400 µl liquid MM in 8-well microscope chambers (Ibidi GmbH) for vegetative growth. Therefore 3000 spores were added and incubated for 18 h at 37°C under illumination. For asexual and sexual investigations, 3000 spores were inoculated on slant agar in petri dishes for 18 h at 37°C. The confocal Axiolab light microscope

was used in combination with the SlideBook 6.0 software package (Intelligent Imaging Innovations GmbH). The calculation of the significance was done with the homepage https://www.graphpad.com (accessed 2021–2023, GraphPad Software) using the unpaired t-test for two groups by using mean, SEM and number of samples. The significance of the ratio of GFP-signals between nucleus and cytoplasm was done with the Wilcoxon–Mann–Whitney test [55].

## Secondary metabolite extraction

Secondary metabolite extraction and LCMS analysis was performed by following the description of [56] and [52]. Therefore strains of interest were inoculated with $5\times10^6$ spores on MM plates for asexual and sexual conditions at 37°C for 5 and 7 days, respectively. Measured data were analyzed using Thermo Scientific Xcalibur 4.1 (Thermo Fisher Scientific) and FreeStyle 1.4 (Thermo Fisher Scientific).

## Crystallization of VeA velvet domain

**Expression and purification of VeA224S.** *E. coli* Rosetta 2 (DE3) cells were transformed with pETM13-VeA[1-224] and grown in ZYM5052 media at 16°C. Cells were lysed in lysis buffer (40 mM HEPES pH 7.4, 40 mM Imidazol and 400 mM NaCl) using a Microfluidics Fluidizer at 0.55 MPa. The lysate was cleared by centrifugation at 30,000 × g for 30 minutes at 4°C. The supernatant was applied to a 5 ml StrepTactinHP column (GE Healthcare) equilibrated with lysis-buffer. After extensive washing the protein was eluted with elution buffer S (lysis-buffer + 2.5 mM des-thiobiotin). The eluate was concentrated to 13.5 mg/ml and used for crystallization.

**Crystallization and data collection.** VeA[1-224] was crystallized by the sitting-drop vapor diffusion method. Rod-shaped crystals of VeA[1-224] grew after 1 day in a condition 25% PEG 3350, 200 mM $(NH_4)_2SO_4$ and 100 mM BisTris/HCl pH 5.5. Crystals were cryo-protected by soaking in reservoir solution supplemented with 12% (v/v) 1,4-butane-diol. X-ray diffraction data were collected at 100K at the Swiss Light Source beamline X06SA. The crystals belong to the spacegroup $P2_12_12$ and have cell dimensions a = 106.17 Å, b = 156.26 Å and c = 52.86 Å, alpha, beta, gamma = 90°.

**Data processing and structure determination.** X-ray diffraction images from the VeA[1-224] crystals were collected at the SLS beamline X06SA and integrated and scaled with the XDS package [57]. Phases were determined by molecular replacement method using a truncated VosA-molecule (PDB ID: 4n6q) as a search model with the program PHASER [58]. A search for four molecules (solvent content 41%) yielded a clear solution. The structure was manually rebuilt in Coot [59] and refined in PHENIX [60] and REFMAC5 [61]. The structure determined at a resolution of 2.4 Å ad refined to an Rwork of 21.93% and Rfree of 25.31% and contains two homodimers in the asymmetric unit. Figures were prepared with PyMOL (Schrödigner LLC). Data collection and refinement statistics are summarized in S6 Table. Atomic coordinates and structure factors have been deposited in the RCSB Protein Data Bank with the accession code: 9I2S.

## Supporting information

**S1 Fig. GFP-signal quantification of truncated *veA* versions and VeA NES/NLS variants relative to unmodified VeA-GFP.** (A) and (B) shows quantification of total GFP-signals as well as the ratio of GFP-signals in cytoplasm (turquoise) and nuclei (green) produced under illuminating conditions inducing asexual and in the dark inducing sexual development (71–108 nuclei for each of three biological replicates). Error bars for quantification represent standard error of the mean (p > 0.05: ns, p ≤ 0.05: *, p ≤ 0.01: **, p ≤ 0.001: ***, p ≤ 0.0001: ****).
(TIF)

**S2 Fig. GFP-signal quantification of VeA NES/NLS variants relative to VeA-GFP with unmodified NES/NLS.** Total GFP-signals as well as the ratio of GFP-signals in in cytoplasm (turquoise) and nuclei (green) produced under vegetative conditions (A) as well as illuminating conditions inducing asexual and in the dark inducing sexual development (B) were

quantified (45–100 nuclei for each of three biological replicates). Error bars for quantification represent standard error of the mean (p > 0.05: ns, p ≤ 0.05: *, p ≤ 0.01: **, p ≤ 0.001: ***, p ≤ 0.0001: ****).
(TIF)

**S3 Fig. Full length VeA is required for correct fungal development.** (A) Asexual development of veA deletion (ΔveA) and complementation (*veA::gfp*) strains as well as truncated versions of VeA (*veA*^Δvelvet*::gfp*, *veA*^46-573*::gfp*, *veA*^1-225*::gfp*, *veA*^46-225*::gfp*) were analyzed after five days incubation at 37°C in light. Plates were incubated in the dark and with limited oxygen for seven days at 37°C for sexual development. The dark brownish color of the veA deletion strain at the bottom of the plate indicates alterations in secondary metabolism combined with failure to form cleistothecia as defects in sexual development. Strains with truncated versions of VeA resembled the veA deletion phenotype to a certain degree. The truncated version having a full or a truncated velvet domain produce more spores and less aerial hyphae compared to the veA deletion phenotype (PMG: photomicrographs, c: cleistothecia, btm: bottom, size bar: 200 μm, cleistothecia 50 μm). (B) Quantification of colony diameters and spore amounts from wildtype, *veA* deletion and *veA* truncations. Colony diameters of strains with truncated VeA have a significantly similar reduction compared to *veA::gfp*. This is also the case for the spore production of mutant strains compared to *veA::gfp*, whereas the tendency shows no significance. In total, this supports that the full VeA protein is required for spore production during asexual development. Quantification results from three biological replicates (3 technical replicates each). Error bars represent standard error of the mean, and the significances were compared to *veA::gfp* complementation strain (p > 0.05: ns, p ≤ 0.0001: ****).
(TIF)

**S4 Fig. The NES regulates VeA protein abundance during asexual development of *A. nidulans*.** Western hybridization of fungal protein extracts from VeA and VeAs complementation strains with amino acid substitutions in NES and NLS motifs were performed. All strains were grown for 20 h at 37°C in liquid minimal medium to enable vegetative growth. The mycelia were then shifted onto 30 ml minimal medium plates for 6–24 h in light at 37°C for asexual development. Samples were collected at identical time points and protein crude extracts were prepared for western hybridization with α-GFP antibody. Signal quantification was performed using BioID software, signals of VeA-GFP (red arrow) were normalized to Ponceau staining. The wild type (wt) VeA protein amount at the 20 h vegetative time point was set to 1 and used to compare the protein amount of following time points of wild type VeA as well as VeA with amino acid substitutions in the NES and NLS motifs. Quantification results from three biological replicates (2 technical replicates each). Error bars for western hybridization represents standard error of the mean the significances were compared to vegetative growth of each strains respectively (p > 0.05: ns, p ≤ 0.05: *, p ≤ 0.01: **, p ≤ 0.001: ***, p ≤ 0.0001: ****).
(TIF)

**S5 Fig. NLS1, NLS2, and NLS3 motifs are conserved in fungal VeA orthologs.** Clustal Omega protein sequence alignments of the three NLS and the NES were performed between fungal VeA orthologs. The NLS1 (blue), NLS2 (turquoise) and NLS3 (dark blue) motifs are conserved in *A. nidulans* (An), *A. fumigatus* (Af), *A. flavus* (Afl), *A. niger* (Ani), *A. oryzae* (Ao) and *N. crassa* (Nc). The NLS3 motif is not conserved in *V. dahliae* (Vd). The NES motif is only conserved in *Aspergillus* species but not in *N. crassa* or *V. dahliae*. Conserved lysine (K), arginine (R), and leucine (L) residues within the consensus sequences are marked in yellow.
(TIF)

**S1 Table. Identified masses and corresponding secondary metabolites.**
(DOCX)

**S2 Table. Plasmids designed and used in this study.**
(DOCX)

**S3 Table. *A. nidulans* strains used in this study.**
(DOCX)

**S4 Table. Primers designed via Lasergene software.**
(DOCX)

**S5 Table. Oligonucleotides used for qRT-PCR.**
(DOCX)

**S6 Table. Data collection and refinement statistics.**
(DOCX)

## Acknowledgments

The authors thank Dr. Christoph Sasse, Dr. Blagovesta Popova and Merle Aden for their technical assistance. Additionally, we would like to acknowledge support and access to the SLS beamline X06SA.

## Author contributions

**Conceptualization:** Anja Strohdiek, Anna M. Köhler, Jennifer Gerke, Gerhard H. Braus.

**Formal analysis:** Anja Strohdiek, Anna M. Köhler, Rebekka Harting, Piotr Neumann.

**Funding acquisition:** Gerhard H. Braus.

**Investigation:** Anja Strohdiek, Anna M. Köhler, Helena Stupperich, Piotr Neumann, Yasar L. Ahmed.

**Resources:** Ralf Ficner, Gerhard H. Braus.

**Supervision:** Ralf Ficner, Gerhard H. Braus.

**Validation:** Anja Strohdiek, Anna M. Köhler, Rebekka Harting, Helena Stupperich, Jennifer Gerke, Emmanouil Bastakis, Piotr Neumann, Yasar L. Ahmed.

**Visualization:** Anja Strohdiek, Rebekka Harting, Piotr Neumann.

**Writing – original draft:** Anja Strohdiek.

**Writing – review & editing:** Anna M. Köhler, Rebekka Harting, Helena Stupperich, Jennifer Gerke, Emmanouil Bastakis, Piotr Neumann, Yasar L. Ahmed, Ralf Ficner, Gerhard H. Braus.

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
