## [Decision Letter · Decision Letter 0]

18 Mar 2025

PGENETICS-D-25-00156

The Aspergillus nidulans velvet domain containing transcription factor VeA is shuttled from cytoplasm into nucleus during vegetative growth and stays there for sexual development, but has to return into cytoplasm for asexual development

PLOS Genetics

Dear Dr. Braus,

Thank you for submitting your manuscript to PLOS Genetics. After careful consideration, we feel that it has merit but does not fully meet PLOS Genetics's publication criteria as it currently stands. Therefore, we invite you to submit a revised version of the manuscript that addresses the points raised during the review process.

Below, we have summarized the reviewers' suggestions. In your revised version, please ensure that all points raised by the reviewers are addressed. Once the revisions are complete, we will be able to consider your manuscript for publication.

-The authors should clearly highlight their new findings and explain how they contribute to understanding fungal development. The abstract should focus more on their experiments rather than sounding like a review.

-The discussion should compare VeA’s role in other fungi, especially regarding its conserved NLS and NES motifs and its involvement in infections.

-The influence of other proteins on VeA’s localization should be briefly discussed, particularly how interactions impact access to NLS and NES motifs.

-Figures and legends need better clarity—mutant and wild-type amino acids should be clearly labeled, molecular weights explained, and experimental conditions properly stated.

-The discussion should address why nuclear localization of VeA does not affect conidia production, despite the expectation that nuclear exclusion is necessary for asexual development.

-Formatting and minor corrections are needed, such as proper gene fusion notation, italicizing species names, and avoiding results in figure legends.

Please submit your revised manuscript within 30 days Apr 17 2025 11:59PM. If you will need more time than this to complete your revisions, please reply to this message or contact the journal office at plosgenetics@plos.org. Please include the following items when submitting your revised manuscript:

We look forward to receiving your revised manuscript.

Kind regards,

Ozgur Bayram

Guest Editor

PLOS Genetics

Giovanni Bosco

Section Editor

PLOS Genetics

Aimée Dudley

Editor-in-Chief

PLOS Genetics

Anne Goriely

Editor-in-Chief

PLOS Genetics

**Journal Requirements:**

- ® on pages: 23, 28, and 29

- TM on pages: 29, and 30.

Potential Copyright Issues:

i) Figures 2(B, C), 3, 4(A, C), 5(A, C)6, 7, S1, S2, S3A, and S4. Please confirm whether you drew the images / clip-art within the figure panels by hand. If you did not draw the images, please provide (a) a link to the source of the images or icons and their license / terms of use; or (b) written permission from the copyright holder to publish the images or icons under our CC BY 4.0 license. Alternatively, you may replace the images with open source alternatives. See these open source resources you may use to replace images / clip-art:

5)  Thank you for stating "The data for the crystal structure is available in the RCSB Protein Data Bank accessable with the code 9I2". Please note that, though access restrictions are acceptable now, your entire minimal dataset will need to be made freely accessible if your manuscript is accepted for publication. This policy applies to all data except where public deposition would breach compliance with the protocol approved by your research ethics board. If you are unable to adhere to our open data policy, please kindly revise your statement to explain your reasoning and we will seek the editor's input on an exemption.

6) Please ensure that the funders and grant numbers match between the Financial Disclosure field and the Funding Information tab in your submission form. Note that the funders must be provided in the same order in both places as well. "Open Access Publication Funds of the University of Göttingen" is missing from the Funding Information tab.

**Reviewers' comments:**

Reviewer's Responses to Questions

Reviewer #1: PGENETICS-D-25-00156

Velvet proteins are key regulators of fungal sexual and asexual development and secondary metabolism. The most well-studied velvet protein is VeA (Velvet A), which plays a crucial role in coordinating fungal growth, reproduction, and the production of secondary metabolites, including toxins and antibiotics. veA protein from Aspergillus nidulans is the most intensively studied one, and the manuscript provides significant answers to the important question about the regulatory role of veA in fungal development. The authors show how a complex shuttle control mechanism for VeA localization either in the nucleus or in cytoplasm is controlling different fungal developmental processes.

The precise nuclear import and export control of velvet proteins is a prerequisite for fungal development and secondary metabolism.

After prediction of the crystal structure of the N-terminal VeA protein, including the velvet domain, the authors provide an extensive functional analysis using different truncated, as well as mutated versions of the veA gene. They use mutants, expressing genes for different NLS, NES variants. For example, NLS motifs are required for asexual development, or for conidiospore or cleistothecia maturation. Similarly, NES allows the coordinated development and secondary metabolism. These analyses are an important part of this manuscript.

My only criticism concerns the designation of the mutant constructs in different analysis. Probably, I had some difficult to follow these investigations, because the NLS and NES mutants are not precisely/unambiguously described. For example, VeA*NES*NLS1*NLS2*NLS3 and the corresponding plasmids are only mentioned at the end of the manuscript (line 687ff). It would be further very helpful to see the different combinations with the amino acid substitutions of the NLS elements (line 88 ff). In Fig. 1A, these substitutions are displayed, however in the text, I miss the corresponding references. An error in Fig. 1A let the reader suggest that two variants are shown for NLS1.

Indeed, the “KR” amino acids are not at position 44/45 (Fig.1A), but instead at position 41/42. This mistake implies that two different AA substitutions were constructed. I suggest mentioning the nomenclatures NLS* etc in Fig. 1A. Table S2 contains the correct positions.

Suggested changes and minor points:

Line 178 “…….addition to the veA:gfp fusion genes with point mutated codons for the signal sequences (Fig 2A).” (I see the mutated codons only in Fig. 1A.)

192 ff: changes and additional combinations with the amino acid substitutions of the NLS elements described before….. (indicate that it is only mentioned in the legend of Fig. 1A.)

Line 63: be more specific “…. fungus starts with asco- or conidiospore germination…..”

Line 426 delete “an”, “Taken together, an all tested…”

Formal comment: Give all species names in the reference titles in italics, e.g. reference 36, 37, 42 (even though the Journal is not consistent in diverse papers).

Overall, this manuscript is a significant contribution to our understanding about the roles of three VeA nuclear localization signals (NLS) and a single nuclear export signal (NES) in fungal development. The manuscript deserves publication in this Journal and is of major importance for geneticists interested in the regulation of fungal, as well as eukaryotic development.

Reviewer #2: The manuscript by Strohdiek et al describes the protein elements that are required for the correct cellular localization of the veA protein, a component of the velvet complex, in the fungus Aspergillus nidulans. The authors identified three nuclear localization signals (NLS) and one nuclear exclusion signal (NES) in VeA, and have created a series of single, and multiple mutations in each of the domains to assay their role in the nuclear localization of the protein during sexual and asexual development, and their role in the accumulation of secondary metabolites. The authors have also characterized the transcription of each mutant gene and the stability of the wild type and mutant proteins during development. The authors have shown that the three NLS domains and the NES act in coordination to regulate the changes in the localization of VeA in the fungal cell during sexual and asexual development. The balance of sexual and asexual development requires a correct cellular localization of VeA and its interaction with other velvet proteins, and the authors have shown that the NLSs and NEs in VeA are required for the correct localization and stability of VeA during the regulation of development and secondary metabolism. The manuscript is well written and the results support the conclusions of the manuscript. I have several comments that I hope will improve the final manuscript.

1. Figure 1 legend. A. "Functional studies of all NLS and the NES sequences were conducted by codon exchanges for the indicated amino acids.". From the figure I think that the authors refer to the amino acids shown below each domain. Please, indicate what are the wt amino acids and the mutant amino acids. Also, explain what are the amino acids shown in the box below the protein scheme. I assume that they are the full amino acid sequences for each domail but I suggest that this is indicated clearly in the legend and in the figure.

2. Results. Page 6. Lines 146-147. “NLS1-3 cooperate in VeA nuclear import and are required to exclude the protein from the nucleus in light when asexual development is promoted”. I suggest that the authors clearly state the growth conditions, including illumination conditions, for the experiments that they plan to perform right at the start of this section. The information is described in materials and methods and also later in the text but I suggest that they provide a clear description at the start for the sake of clarity. Sometimes the authors refer to growth conditions as under light or dark, in others, they specify the growth conditions are promoting sexual versus asexual development. Also the illumination conditions in vegetative growth should be described as well.

3. Figure 2 legend and figures. A. Please, in the legend replace “Schema” by “Scheme”. Please, indicate in the figure that the numbers above the first scheme correspond to the amino acids of the protein. In the mRNA, please, indicate in the mRNA scheme the presence of an intron. In the figure, the kDa above the VeA delta velvet presumably indicate the molecular weight of each segment of the fused protein. Please, indicate the kDa for the other protein fusions. If the authors consider that this is too much information for the figure, please, consider deleting the MW for this specific fusion. B. Please, indicate when the cultures were kept in light or in dark with L or D in the figure, in addition to indicating the vegetative or developmental program. C. Please, indicate in the legend the conditions for vegetative growth (liquid culture with shaking?, dark or light?).

4. Results. Page 7. Line 169. “increased nuclear accumulation in light of approximately 30%...” Since the quantifications are shown in fig. S1 I suggest indicating this supplementary figure after this statement.

5. Results. Page 7. Line 178. Gene fusions are usually shown with the symbol “::” between the two fused segments. Please, correct.

6. Legend to figure S1. Lines 981-983. Please correct GPF to GFP and delete the double “in”.

7. Legend to figure 4. Please, do not include results in the legend (for example, lines 260-266). Idem for legend to figure 5.

8. Discussion. Mutations in NLS and/or NES result in nuclear localization of VeA during asexual growth (light), however, the amount of conidia in these mutant strains is similar to that obtained from the wild type (figures 4-5). The authors should discuss how these contradictory results could be explained under the hypothesis that VeA should be excluded from the nuclei to allow asexual development.

9. Discussion. Can the results obtained in this manuscript be extended to the activity of other VeA homologs? I think that the text will improve if the authors could include a discussion on the presence of NLS or NES in other VeA homologs, at least in the homologs that have been described in more detail such as in N. crassa or Fusarium. Also, I suggest that the authors could incorporate in the discussion what could be consequence of the export of VeA from the nuclei in the interaction of VeA with the other velvet proteins and the formation of regulatory complexes. Could some of the phenotypes observed be a consequence of diferential interaction with other velvet proteins in the NLS or NES mutants?

Reviewer #3: Review of the manuscript PGENETICS-D-25-00156

The heterotrimeric velvet-complex VelB/VeA/LaeA is the central regulator of sexual and asexual morphogenesis and secondary metabolism in fungi. In Aspergillus nidulans, the localisation of the transcription factor VeA depends on light. In the dark, VeA is imported to the nucleus to allow sexual development, while light induces cytosolic localisation and the asexual pathway. The authors investigated in their work the role of the three nuclear localisation signals (NLS1-NLS3) and the nuclear export sequence (NES) for the localisation of VeA. This was achieved by the construction of various VeA-GFP fusion proteins with point mutations in NLS1, NLS2, NLS3 and NES. In addition, truncated versions of VeA fused to GFP were constructed. These experiments revealed that VeA is already localised in the nucleus during vegetative growth. Furthermore, the combination of NLS1, NLS2 and NES motifs seems to be required for the nuclear import of VeA. In contrast, NLS3 probably cooperates with NES to induce cytoplasmic localisation and asexual spore production. Cytoplasmic localisation of VeA seems to prevent its degradation and stabilises the protein. The authors also determined the crystal structure of the N-terminal part of VeA which illustrated the spatial localisation of the three NLS and the NES-motif. In addition, the authors showed that the transition to either sexual or asexual development in dependence of the VeA localisation went along with the production of specific secondary metabolites, which was detected by LC-MS-analysis. In summary, the manuscript demonstrates the complex regulation of the VeA localisation for coordinating fungal development

The topic of the manuscript addresses a fundamental and highly relevant research in the field of fungal biology and is based on well performed experiments. The number of constructed A. nidulans strains is impressive and the findings are very interesting. There are few points I recommend the authors to address before a possible publication of the manuscript:

1) The authors should emphasise more strongly which of their findings are new and how they can complete the picture of developmental regulation in fungi. For example, the abstract needs improvement to summarise the author’s experimental work and reads more like an abstract for a review article.

2) VeA orthologues exist in many fungi, for some it was shown that they are either required for infections of plants or humans, e.g. Aspergillus, Fusarium, Cochliobolus and Magnaporthe. Can the author’s findings be discussed in the context of other fungi? How conserved are the NLS and NES motifs in such fungi? The authors should address these aspects in their manuscript.

3) The localisation of VeA is also influenced by its binding to other proteins. For example, importin KapA recognises the VeA-VelB dimer for nuclear import or the LeaA-like methyltransferase LlmF has an impact on the localisation of VeA. I recommend the authors shortly discuss how the interaction of VeA with other proteins could influence the accessibility to the NLS or NES motifs.

Minor points

p. 11, line 290

Shortly mention the amino acid substitutions.

p. 23, line 582

Windows error message should be deleted.

P. 28, line 706

What means without air? Please specify.

p. 29, line 722

Did the authors use the Ponceau signal to normalise the Western blot data? More details should be added. Can the authors exclude saturation of the Ponceau signal?

**Have all data underlying the figures and results presented in the manuscript been provided?**

Reviewer #1: Yes

Reviewer #2: Yes

Reviewer #3: Yes

PLOS authors have the option to publish the peer review history of their article (what does this mean? ). If published, this will include your full peer review and any attached files.

**Do you want your identity to be public for this peer review?** For information about this choice, including consent withdrawal, please see our Privacy Policy .

Reviewer #1: No

Reviewer #2: No

Reviewer #3: No

**Figure resubmission:**
---

## [Decision Letter · Decision Letter 1]

10 Apr 2025

Dear Dr Braus,

We are pleased to inform you that your manuscript entitled "The Aspergillus nidulans velvet domain containing transcription factor VeA is shuttled from cytoplasm into nucleus during vegetative growth and stays there for sexual development, but has to return into cytoplasm for asexual development" has been editorially accepted for publication in PLOS Genetics. Congratulations!

Yours sincerely,

Ozgur Bayram

Guest Editor

PLOS Genetics

Giovanni Bosco

Section Editor

PLOS Genetics

Aimée Dudley

Editor-in-Chief

PLOS Genetics

Anne Goriely

Editor-in-Chief

PLOS Genetics

Comments from the reviewers (if applicable):

Reviewer's Responses to Questions

**Comments to the Authors:**

Reviewer #1: Dear authors,

Thank you for carefully considering the comments by the reviewers.

Reviewer #2: The authors have addressed all the questions that I made. I have no other objections to the manuscript.

Reviewer #3: All critical point raised by the reviewers have been addressed.

**Have all data underlying the figures and results presented in the manuscript been provided?**

Reviewer #1: Yes

Reviewer #2: Yes

Reviewer #3: Yes

PLOS authors have the option to publish the peer review history of their article (what does this mean? ). If published, this will include your full peer review and any attached files.

**Do you want your identity to be public for this peer review?** For information about this choice, including consent withdrawal, please see our Privacy Policy .

Reviewer #1: No

Reviewer #2: No

Reviewer #3: No

**Data Deposition**

http://datadryad.org/submit?journalID=pgenetics&manu=PGENETICS-D-25-00156R1

**Press Queries**

---

## [Editor Report · Acceptance letter]

PGENETICS-D-25-00156R1

The Aspergillus nidulans velvet domain containing transcription factor VeA is shuttled from cytoplasm into nucleus during vegetative growth and stays there for sexual development, but has to return into cytoplasm for asexual development

Dear Dr Braus,

We are pleased to inform you that your manuscript entitled "The Aspergillus nidulans velvet domain containing transcription factor VeA is shuttled from cytoplasm into nucleus during vegetative growth and stays there for sexual development, but has to return into cytoplasm for asexual development" has been formally accepted for publication in PLOS Genetics! Your manuscript is now with our production department and you will be notified of the publication date in due course.

With kind regards,

Anita Estes

PLOS Genetics

On behalf of:
